

# DOG1 is commonly expressed in pancreatic adenocarcinoma but unrelated to cancer aggressiveness

Kristina Jansen[1]   Franziska Büscheck[1]   Katharina Moeller[1]   Martina Kluth[1]   Claudia Hube-Magg[1]   Niclas Christian Blessin[1]   Daniel Perez[1]   Jakob Izbicki[1]   Michael N[...]   Hamid Mofid[3]   Thies Daniels[4]   Ulf Nahrstedt[5]   Christoph Fraune[1]   Frank Jacobsen[1]   Christian Bernreuther[1]   Patrick Lebok[1]   Guido Sauter[1]   Ria Uhlig[1]   Waldemar Wilczak[1]   Ronald Simon[1]   Stefan Steurer[1]   Eike Burandt[1]   Andreas Marx[1,6]   Till Krech[1,7]   Till Clauditz[1]

[1] University Medical Center Hamburg-Eppendorf, Hamburg, Germany
[2] Itzehoe Medical Center, Itzehoe, Germany
[3] Regio Clinic Pinneberg, Pinneberg, Germany
[4] Albertinen Hospital, Hamburg, Germany
[5] Schoen Clinic Hamburg Eilbek, Hamburg, Germany
[6] Academic Hospital Fuerth, Fuerth, Germany
[7] Institute of Pathology, Clinical Center Osnabrueck, Osnabrück, Germany

Corresponding author
Ronald Simon, r.simon@uke.de

## ABSTRACT

**Background**. DOG1 (ANO1; TMEM16A) is a voltage-gated calcium-activated chloride and bicarbonate channel. DOG1 is physiologically expressed in Cajal cells, where it plays an important role in regulating intestinal motility and its expression is a diagnostic hallmark of gastrointestinal stromal tumors (GIST). Data on a possible role of DOG1 in pancreatic cancer are rare and controversial. The aim of our study was to clarify the prevalence of DOG1 expression in pancreatic cancer and to study its association with parameters of cancer aggressiveness.

**Methods**. DOG1 expression was analyzed by immunohistochemistry in 599 pancreatic cancers in a tissue microarray format and in 12 cases of pancreatitis on large tissue sections.

**Results**. DOG1 expression was always absent in normal pancreas but a focal weak expression was seen in four of 12 cases of pancreatitis. DOG1 expression was, however, common in pancreatic cancer. Membranous and cytoplasmic DOG1 expression in tumor cells was highest in pancreatic ductal adenocarcinomas (61% of 444 interpretable cases), followed by cancers of the ampulla Vateri (43% of 51 interpretable cases), and absent in 6 acinus cell carcinomas. DOG1 expression in tumor associated stroma cells was seen in 76 of 444 (17%) pancreatic ductal adenocarcinomas and in seven of 51 (14%) cancers of the ampulla Vateri. Both tumoral and stromal DOG1 expression were unrelated to tumor stage, grade, lymph node and distant metastasis, mismatch repair protein deficiency and the density of CD8 positive cytotoxic T-lymphocytes in the subgroups of ductal adenocarcinomas and cancers of ampulla Vateri. Overall, the results of our study indicate that DOG1 may represent a potential biomarker for pancreatic cancer diagnosis and a putative therapeutic target in pancreatic cancer. However, DOG1 expression is unrelated to pancreatic cancer aggressiveness.

# INTRODUCTION

Pancreatic cancer is the 11th most frequent cancer, but the 2nd leading cause of cancer-related mortality in the United States of America (*Global Cancer Observatory, 2020*). Both the incidence and mortality are continuously rising (*Rahib et al., 2014*). The poor prognosis of pancreatic cancer results from the scarcity of early symptoms and consecutively a late diagnosis of locally advanced or metastatic cancers in most patients. The 5-year survival independent of the tumor stage remains at 8% (*Siegel, Miller & Jemal, 2017*). The histopathological diagnosis of pancreatic cancer can be challenging due to the often limited quantity of cells or tissue obtainable by fine needle aspiration/biopsy and considerable inflammation in the surroundings of these cancers (*Mangiavillano et al., 2020*).

DOG1 (Discovered On Gastrointestinal Stromal Tumors Protein 1), also known as Transmembrane Protein 16A (TMEM16A) or Anoctamin-1 (ANO1) is a voltage-gated calcium-activated chloride and bicarbonate channel (*Caputo et al., 2008*; *Yang et al., 2008*). DOG1 is highly expressed in the gastrointestinal interstitial cells of Cajal, where it plays an important role in epithelial chloride secretion mediating intestinal motility (*Miettinen, Wang & Lasota, 2009*; *Chevalier et al., 2020*). Calcium-activated chloride channel blocking drugs like niflumic acid have been shown to block slow waves (pacemaker activity)—which produce motility—in the human small intestine and stomach (*Hwang et al., 2009*). High levels of DOG1 expression are a diagnostic hallmark of gastrointestinal stromal tumor (GIST), a tumor derived from these cells (*Miettinen, Wang & Lasota, 2009*; *West et al., 2004*; *Kindblom et al., 1998*; *Sircar et al., 1999*). However, DOG1 expression was also reported to occur in squamous cell carcinomas of the esophagus and head and neck, thyroid carcinomas, and adenocarcinomas of endometrium, stomach and colon (*Miettinen, Wang & Lasota, 2009*; *Friedrich et al., 2016*; *Chenevert et al., 2012*; *Hemminger & Iwenofu, 2012*; *Yu et al., 2019*). Data on the prevalence and significance of DOG1 expression in pancreatic cancer are limited and partly controversial. In a study employing immunohistochemistry (IHC) on a tissue microarray (TMA), *Hemminger et al. (2012)* identified patchy low to moderate intensity DOG1 immunostaining in 8 (7%) of 112 pancreatic adenocarcinomas. In contrast, *Crottes et al. (2019)* scrutinized available databases for signs of up-regulation of DOG1 protein and mRNA in pancreatic adenocarcinoma and determined that DOG1 is up-regulated in 75% of pancreatic carcinomas. From data derived from the TCGA database, these authors further concluded that high levels of DOG1 were correlated with low patient survival probability. Evidence for a functionally active role of DOG1 in pancreatic cancer cells come from four previous studies showing elevated DOG1 expression in pancreatic cancer cell lines and demonstrating that inhibition, knockdown or knockout of DOG1 attenuates cell motility, migration, and proliferation and promotes cell cycle arrest in G0/G1 phase *in vitro* and *in vivo* (*Crottes et al., 2019*; *Sauter et al., 2015*; *Stanich et al., 2011*; *Mazzone et al., 2012*). *Nielsen, Mortensen & Detlefsen (2018)* described DOG1 expression to

also occur in pancreatic cancer associated fibroblasts. In a thorough study analyzing DOG1 expression in well-defined stroma compartments of eight pancreatic adenocarcinomas they found a higher rate of DOG1 expression in cancer-associated fibroblasts (CAFs) that were located in the immediate neighborhood of cancer cells (juxtatumoral stroma) than in the cancer periphery (peritumoral stroma).

The aims of this study were to clarify the prevalence of DOG1 expression in epithelial and stromal cells of pancreatic carcinomas and to identify potential associations with parameters of cancer aggressiveness. For this purpose, a cohort of 599 pancreatic carcinomas was analyzed for DOG1 expression by IHC in a TMA format.

## MATERIALS & METHODS

### Tissue microarray

The 599 samples were diagnosed at the Institute of Pathology, University Medical Center Hamburg-Eppendorf, Hamburg, Germany: 532 ductal adenocarcinomas, 61 adenocarcinomas of the ampulla Vateri, and 6 acinar cell carcinomas. The TMA was made as described in *Kononen et al. (1998)*. Clinico-pathological parameters were obtained from the pathology reports (Table 1). The molecular database attached to the TMA contained results on mismatch repair deficiency (dMMR, surrogate for microsatellite instability, MSI) measured by MLH1, MSH2, PMS2, and MSH6 immunohistochemistry in 519 cases from a previous study (*Fraune et al., 2020*) and the density of CD8 positive cytotoxic T-lymphocytes in 599 cases (*Blessin et al., 2020*). Large sections from 12 pancreatectomy specimens from patients with pancreatitis not suffering from carcinoma were also analyzed. The use of archived material for research purpose as well as patient data analysis has been approved by local laws (HmbKHG, §12) and by the ethics committee of Hamburg (WF-049/09). The work was done in compliance with the Helsinki Declaration.

### Immunohistochemistry

TMA sections were freshly cut, processed and stained the same day. Slides were deparaffinized with xylol, rehydrated and exposed to heat-induced antigen retrieval. Endogenous peroxidase activity was blocked with Dako Peroxidase Blocking Solution (#52023; Agilent, Santa Clara, CA, USA) for 10 min. Anti-DOG1 mouse monoclonal antibody MSVA-201M (MS Validated Antibodies, Hamburg, Germany) was applied at 37 °C, 60 min, at 1:150. Staining was visualized with the EnVision Kit (#K5007; Agilent, Santa Clara, CA, USA) and counterstained with Haemalaun. DOG1 staining was predominantly membranous in pancreatic cancer cells. Scoring of the staining intensity was semi-quantitatively assessed as previously described in *Juhnke et al. (2017)*. Specifically, four grades were defined: *Negative* (no staining at all), *weak* (staining intensity of 1+ in ≤ 70% of the tumor cells or staining intensity of 2+ in ≤ 30% of the tumor cells), *moderate* (staining intensity of 1+ in >70% of the tumor cells, staining intensity of 2+ in >30% but in ≤ 70% of the tumor cells or staining intensity of 3+ in ≤ 30% of the tumor cells), *strong* (staining intensity of 2+ in >70% of the tumor cells or staining intensity of 3+ in >30% of the tumor cells).

| Table 1 Patient cohort. | |
| --- | --- |
| | all tumors ($n = 599$) |
| **Tumor type** | |
| ductal adenocarcinoma | 532 (89%) |
| acinar cell carcinoma | 6 (1%) |
| adenocarcinoma of the ampulla Vaterii | 61 (10%) |
| **Tumor stage** | |
| pT1 | 20 (3%) |
| pT2 | 93 (16%) |
| pT3 | 435 (73%) |
| pT4 | 49 (8%) |
| **Grade** | |
| 1 | 19 (3%) |
| 2 | 420 (74%) |
| 3 | 130 (23%) |
| **Lymph node status** | |
| pN0 | 135 (23%) |
| pN+ | 461 (77%) |
| **Distant metastasis** | |
| pM0 | 474 (79%) |
| pM1 | 123 (21%) |
| **Surgical margin status** | |
| R0 | 324 (58%) |
| R1 | 231 (42%) |

## Statistics

JMP® software (SAS Institute Inc., Cary, NC, USA) was used for contingency tables and chi$^2$-tests to search for associations between DOG1 expression and histological subtypes, clinico-pathological parameters and dMMR. A $p$-value $\leq 0.05$ was considered as statistically significant.

## RESULTS

### Technical issues

On our TMA, 501 of 599 (83.6%) pancreatic cancers were analyzable in the DOG1 IHC analysis. Reasons for non-informative cases ($n = 98$, 16.4%) included lack of tissue samples or absence of unequivocal cancer tissue in the TMA spot.

### DOG1 expression in pancreatic cancers

DOG1 immunostaining could be observed in both cancer cells and in tumor associated stromal cells. A predominantly membranous cancer cell staining was seen in 294 (58.7%) of the 501 interpretable cancers. The observed staining pattern were variable and ranged from focal staining of various intensity to intense diffuse positivity. Stroma cell staining was often periglandular or "juxtatumoral" but did also involve cells that were more remote from

cancer cells. Representative images are shown in Fig. 1. The frequency of DOG1 positive cancer cell staining was highest in ductal adenocarcinomas (61.3%; $n = 444$), followed by adenocarcinomas of the ampulla of Vateri (43.1%; $n = 51$). DOG1 positive stromal cells were found in 76 of 444 (17.1%) ductal adenocarcinomas, and 7 of 51 (13.7%) adenocarcinomas of the ampulla of Vateri. DOG1 immunostaining both tumoral and stromal was absent in acinar cell carcinoma ($n = 6$). DOG1 staining was also completely absent in normal pancreatic cells. A weak to moderate focal DOG1 staining was seen, however, in 4 of 12 large sections of pancreatitis cases. Statistical associations were not observed between both stromal and tumor cell DOG1 staining and clinico-pathological parameters in the analysis of ductal adenocarcinomas of the pancreas ($p > 0.1$ each; Tables 2 and 3) and of cancers of the ampulla Vateri ($p > 0.5$ each except of pT with $p = 0.0104$; Tables 2 and 3). DOG1 staining was also unrelated to dMMR and the density of CD8 positive lymphocytes in ductal adenocarcinomas and adenocarcinomas of the ampulla of Vateri ($p \geq 0.1$; Tables 2–4).

## DISCUSSION

The results of our study demonstrate that tumoral DOG1 expression is frequent in both ductal adenocarcinoma of the pancreas (61.3% of 444 cancers) and adenocarcinomas of the ampulla Vateri (43.1% of 51 cancers). This observation is consistent with data derived from large databases on RNA and protein expression in cancers. Crottes et al. (2019) had scrutinized available databases for signs of up-regulation of DOG1 mRNA and protein expression in pancreatic adenocarcinoma and determined that DOG1 was up-regulated in 75% of pancreatic cancers. Our frequency of tumoral DOG1 immunostaining in pancreatic cancer is markedly higher than in the only previous study investigating DOG1 expression by IHC. In a study analyzing 112 pancreatic adenocarcinomas in a TMA format, Hemminger et al. (2012) found a weak to moderate DOG1 positivity in 8 (7%) of all carcinomas. The use of different antibodies, staining protocols, and criteria for defining positivity are the most common causes for different outcomes of studies employing IHC (Janardhan et al., 2018). In addition, in case of TMA studies, the time span between the cutting of the TMA section and its immunohistochemical staining has a marked impact on the staining intensity of many antibodies. Even a slide age of two weeks can lead to a marked reduction of staining (Mirlacher et al., 2004).

Beside the expression of DOG1 in tumor cells, Nielsen et al. (2020) have recently shown that DOG1 can also be expressed in CAFs. In their study, Nielsen et al. focused on the analysis of the tumor microenvironment—especially the juxtatumoral, peripheral, lobular, septal, peripancreatic, and regressive stroma compartments—of chemotherapy-naïve (CTN-PC; $n = 10$) and neoadjuvant treated (NAT-PC; $n = 10$) pancreatic adenocarcinomas and found that DOG1 was particularly overexpressed in CAFs that were located in close contact to cancer cells (juxtatumoral CAFs). The authors concluded from their data that juxtatumoral CAFs characterized by strong DOG1 expression and several other markers might promote the proliferation and invasion of cancer cells. A pathogenetic role of DOG1 expression in CAFs is potentially also supported by our data, as DOG1 expressing fibroblasts

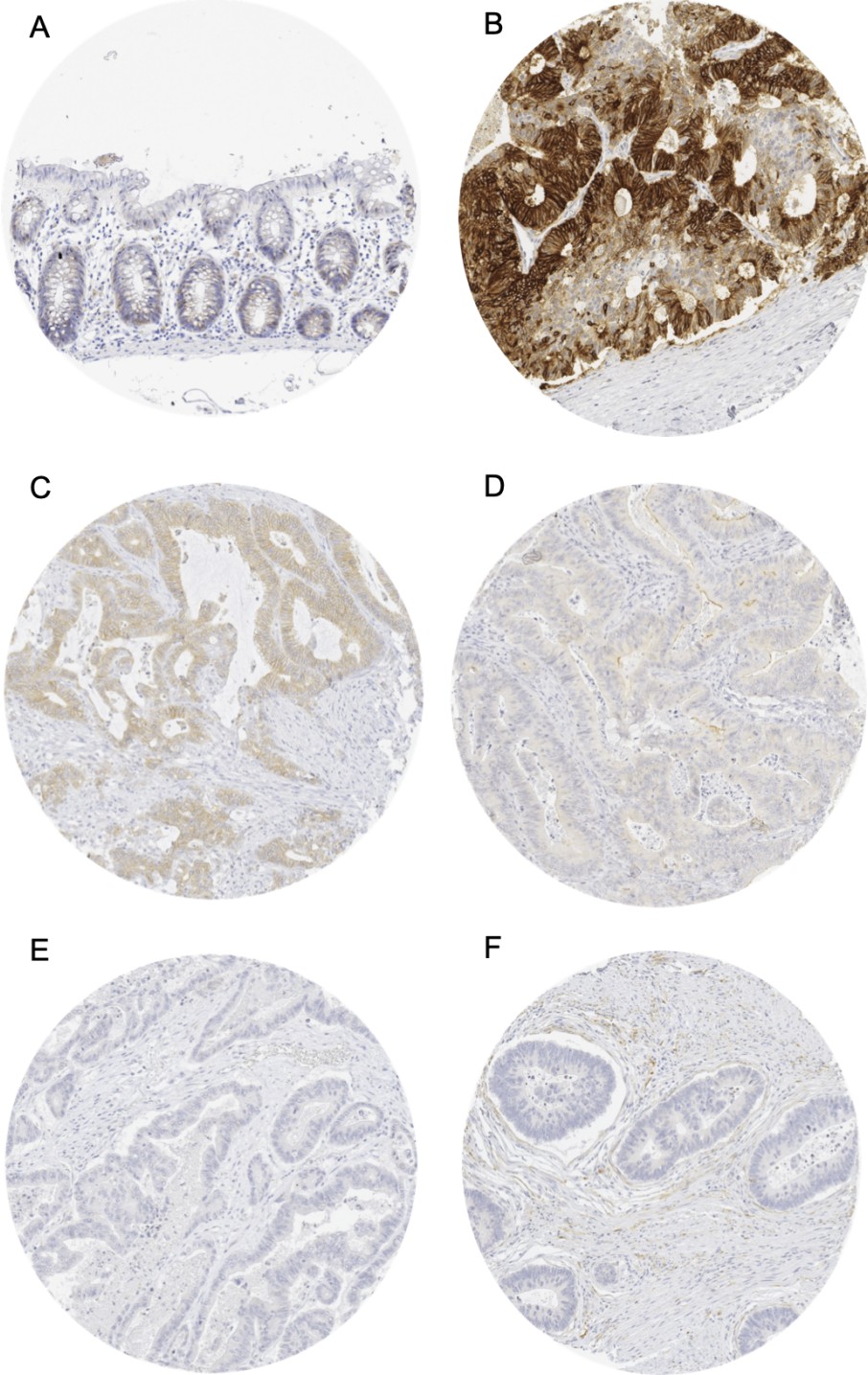

**Figure 1 DOG1 immunostaining.** (A–D) DOG1 positive ductal adenocarcinomas with strong (A), moderate (B), focal moderate (C), and weak immunostaining of tumor cells (D). (A) and (C) also contain DOG1 negative normal ducts (arrow). (E & F) DOG1 negative cancers with diffuse (E) and periductal DOG1 (F) staining of stromal cells (F).

**Table 2 DOG1 immunostaining in cancer cells and cancer phenotype.**

| | | | DOG1 immunostaining in cancer cells | | | | |
|---|---|---|---|---|---|---|---|
| | | n evaluable | Negative (%) | Weak (%) | Moderate (%) | Strong (%) | *p* value |
| Ductal adenocarcinomas | | 444 | 38.7 | 26.1 | 17.8 | 17.3 | |
| Tumor stage | pT1 | 12 | 16.7 | 25 | 25 | 33.3 | |
| | pT2 | 62 | 35.5 | 27.4 | 22.6 | 14.5 | 0.5262 |
| | pT3-4 | 368 | 39.7 | 26.1 | 16.8 | 17.4 | |
| Grade | 1 | 11 | 36.4 | 18.2 | 18.2 | 27.3 | |
| | 2 | 317 | 40.1 | 25.2 | 18 | 16.7 | 0.4193 |
| | 3 | 94 | 27.7 | 31.9 | 19.1 | 21.3 | |
| Lymph node status | pN0 | 89 | 44.9 | 23.6 | 15.7 | 15.7 | 0.5912 |
| | pN+ | 352 | 36.9 | 26.7 | 18.5 | 17.9 | |
| Surgical margin status | R0 | 219 | 37.4 | 23.3 | 19.2 | 20.1 | 0.1260 |
| | R1 | 192 | 37.5 | 32.3 | 16.1 | 14.1 | |
| Mismatch repair | intact | 400 | 39.3 | 26 | 17 | 17.8 | 0.5187 |
| | deficent | 3 | 33.3 | 0 | 33.3 | 33.3 | |
| Adenocarcinoma of the ampulla Vaterii | | 51 | 56.9 | 17.6 | 13.7 | 11.8 | |
| Tumor stage | pT1 | 3 | 0 | 66.7 | 0 | 33.3 | |
| | pT2 | 9 | 22.2 | 22.2 | 44.4 | 11.1 | 0.0104 |
| | pT3-4 | 39 | 69.2 | 12.8 | 7.7 | 10.3 | |
| Grade | 1 | 0 | – | – | – | – | |
| | 2 | 33 | 54.5 | 18.2 | 18.2 | 9.1 | 0.5299 |
| | 3 | 18 | 61.1 | 16.7 | 5.6 | 16.7 | |
| Lymph node status | pN0 | 11 | 45.5 | 18.2 | 36.4 | 0 | 0.0595 |
| | pN+ | 40 | 60 | 17.5 | 7.5 | 15 | |
| Surgical margin status | R0 | 43 | 55.8 | 14 | 16.3 | 14 | 0.1841 |
| | R1 | 7 | 71.4 | 28.6 | 0 | 0 | |
| Mismatch repair | intact | 48 | 56.3 | 18.8 | 12.5 | 12.5 | – |
| | deficent | 0 | – | – | – | – | |

were seen in 17% of pancreatic adenocarcinomas but neither in normal pancreas nor in chronic pancreatitis.

That DOG1 expression in tumor or stromal cells did not show any associations with tumor stage, grade, or nodal and distant metastasis in our study argues against a clinically significant impact of DOG1 expression on pancreatic cancer aggressiveness. This is in contrast with data derived from Crottès et al. from the TCGA database, where high levels of TMEM16A were linked to a low patient survival probability in a cohort of 146 patients (*Crottes et al., 2019*). Comparing protein expression measured by IHC and RNA expression data is particularly challenging in pancreatic cancer because of the high average stroma content of this tumor (*Feig et al., 2012*). RNA data may therefore be highly diluted in
**Table 3   DOG1 immunostaining in stroma cells and cancer phenotype.**

| | | DOG1 immunostaining in stroma cells | | | |
|---|---|---|---|---|---|
| | | n evaluable | Negative (%) | Positive (%) | *p* value |
| **Ductal adenocarcinomas** | | 444 | 82.9 | 17.1 | |
| **Tumor stage** | pT1 | 12 | 91.7 | 8.3 | |
| | pT2 | 62 | 87.1 | 12.9 | 0.4117 |
| | pT3-4 | 368 | 82.1 | 17.9 | |
| **Grade** | 1 | 11 | 81.8 | 18.2 | |
| | 2 | 317 | 83.6 | 16.4 | 0.8236 |
| | 3 | 94 | 80.9 | 19.2 | |
| **Lymph node status** | pN0 | 89 | 82.0 | 18.0 | |
| | pN+ | 352 | 83.0 | 17.1 | 0.8359 |
| **Surgical margin status** | R0 | 219 | 81.3 | 18.7 | |
| | R1 | 192 | 84.4 | 15.6 | 0.4064 |
| **Mismatch repair** | intact | 400 | 83.2 | 16.8 | |
| | deficent | 3 | 66.7 | 33.3 | 0.4885 |
| **Adenocarcinoma of the ampulla Vaterii** | | 51 | 86.3 | 13.7 | |
| **Tumor stage** | pT1 | 3 | 66.7 | 33.3 | |
| | pT2 | 9 | 100.0 | 0.0 | 0.1748 |
| | pT3-4 | 39 | 84.6 | 15.4 | |
| **Grade** | 1 | 0 | – | – | |
| | 2 | 33 | 81.8 | 18.2 | 0.1825 |
| | 3 | 18 | 94.4 | 5.6 | |
| **Lymph node status** | pN0 | 11 | 81.8 | 18.2 | |
| | pN+ | 40 | 87.5 | 12.5 | 0.6374 |
| **Surgical margin status** | R0 | 43 | 86.1 | 14.0 | |
| | R1 | 7 | 85.7 | 14.3 | 0.9813 |
| **Mismatch repair** | intact | 48 | 84.8 | 15.2 | – |
| | deficent | 0 | – | – | |

many of pancreatic cancers and partly reflect tumor cell density. Moreover, if cancers are preselected for high tumor cell content, a selection bias may apply. It may well be that associations between molecular drivers of cancer aggressiveness and unfavorable tumor features are—due to their overall very poor prognosis—particularly difficult to identify in pancreatic cancer. That the density of CD8 positive cytotoxic T-lymphocytes did not vary between tumors expressing different levels of DOG1 argues against a particular role
**Table 4   DOG1 immunostaining and CD8 positivity.**

| | DOG1 in cancer cells | n evaluable | CD8 density (cells/mm²) | p value |
|---|---|---|---|---|
| Ductal adenocarcinomas | negative | 172 | 239.6 ± 21.9 | |
| | weak | 116 | 226.0 ± 26.7 | 0.7658 |
| | moderate | 79 | 197.8 ± 32.4 | |
| | strong | 77 | 226.8 ± 32.8 | |
| Adenocarcinoma of the ampulla Vaterii | negative | 29 | 291.1 ± 78.8 | |
| | weak | 9 | 121.7 ± 141.5 | 0.1048 |
| | moderate | 7 | 637.1 ± 160.5 | |
| | strong | 6 | 179.6 ± 173.3 | |
| | **DOG1 in stroma cells** | **n evaluable** | **CD8 density (cells/mm²)** | **p value** |
| Ductal adenocarcinomas | negative | 368 | 227.7 ± 15.0 | 0.8326 |
| | positive | 76 | 220.1 ± 33.0 | |
| Adenocarcinoma of the ampulla Vaterii | negative | 44 | 318.4 ± 66.3 | 0.3577 |
| | positive | 7 | 152.3 ± 166.2 | |

of DOG1 in the extent of tumor immunogenicity or pathways with a function in immune response.

Due to the general role of DOG1 overexpression in tumorigenesis and progression as well as the absence or low level of DOG1 expression in most normal tissues, DOG1 may also represent a suitable drug target. Studies have shown that partial or total inhibition of DOG1 with T16Ainh-A01 and CaCCinh-A01 leads to reduced channel activity, cell viability, cell proliferation, cell migration, increased apoptosis, and cell cycle arrest in G0/G1 phase in GIST and cancer cells of the breast, bladder, head and neck, and esophagus *in vitro* (*Frobom et al., 2019*; *Guan et al., 2016*; *Berglund et al., 2014*; *Duvvuri et al., 2012*; *Britschgi et al., 2013*) and reduced tumor growth of lung, breast, and head and neck carcinomas *in vivo* (*Hu, Zhang & Jiang, 2019*; *Kulkarni et al., 2017*). In addition, three studies showed that combined inhibition of DOG1 and EGFR or DOG1 and HER2 leads to reduced cell growth in a cooperative manner and that DOG1 inhibition can reverse resistance to EGFR or HER2 therapies *in vitro* and *in vivo* (*Kulkarni et al., 2017*; *Fujimoto et al., 2017*; *Bill et al., 2015*). Overall, this shows that DOG1 is a promising candidate for a new target cancer therapy. Especially in pancreatic cancer, this deadly tumor entity with only a few therapy options and a high frequency of DOG1 expression.

The data from this study also suggest a potential diagnostic utility of DOG1 IHC in pancreatic cancer. Since DOG1 expression was detectable in more than 60% of pancreatic adenocarcinomas and >40% of adenocarcinomas of the ampulla Vateri, but completely absent in normal pancreatic tissues, a positive DOG1 immunostaining in a pancreatic biopsy may serve as an argument for malignancy. It is of note, however, that DOG1 expression alone cannot secure a diagnosis of pancreatic cancer as weak to moderate focal DOG1 immunostaining was also observed in four of 12 pancreatitis specimens in our study. Current panels that are used to support the diagnosis of malignancy in pancreatic

biopsies typically include CA19-9, CK7, CK19, MUC1, and CEA, (*Wong & Chu, 2012*) and could be complemented by DOG1.

## CONCLUSIONS

In summary, the results of this study show that DOG1 is frequently expressed in pancreatic adenocarcinoma. Although DOG1 expression is unrelated to parameters of cancer aggressiveness, it may be a suitable diagnostic marker and an excellent therapeutic target in pancreatic cancer.

## ACKNOWLEDGEMENTS

We are grateful to Melanie Witt, Inge Brandt, Maren Eisenberg, and Sünje Seekamp for excellent technical assistance.

### Funding
The authors received no funding for this work.

### Competing Interests
The DOG1 antibody clone MSVA-201M was received from MS Validated Antibodies GmbH (owned by a family member of Guido Sauter).

### Author Contributions
- Kristina Jansen, Ronald Simon and Till Clauditz conceived and designed the experiments, performed the experiments, analyzed the data, prepared figures and/or tables, authored or reviewed drafts of the paper, and approved the final draft.
- Franziska Büscheck, Katharina Moeller, Martina Kluth, Claudia Hube-Magg, Niclas Christian Blessin, Christoph Fraune, Frank Jacobsen, Christian Bernreuther and Patrick Lebok conceived and designed the experiments, performed the experiments, analyzed the data, prepared figures and/or tables, and approved the final draft.
- Daniel Perez, Jakob Izbicki, Michael Neipp, Hamid Mofid, Thies Daniels and Ulf Nahrstedt conceived and designed the experiments, performed the experiments, prepared figures and/or tables, collection of samples, and approved the final draft.
- Guido Sauter and Waldemar Wilczak conceived and designed the experiments, performed the experiments, prepared figures and/or tables, authored or reviewed drafts of the paper, and approved the final draft.
- Ria Uhlig performed the experiments, analyzed the data, prepared figures and/or tables, and approved the final draft.
- Stefan Steurer and Eike Burandt conceived and designed the experiments, performed the experiments, analyzed the data, prepared figures and/or tables, and approved the final draft.

- Andreas Marx and Till Krech conceived and designed the experiments, performed the experiments, analyzed the data, prepared figures and/or tables, collection of samples, and approved the final draft.

### Human Ethics

The following information was supplied relating to ethical approvals (i.e., approving body and any reference numbers):

The use of archived remnants of diagnostic tissues for manufacturing of tissue microarrays and their analysis for research purpose as well as patient data analysis has been approved by local laws (HmbKHG, §12) and by the local ethics committee (Ethics Commission Hamburg, WF-049/09). All work has been carried out in compliance with the Helsinki Declaration.

### Data Availability

The raw data of DOG1 immunostaining obtained from the pancreatic cancer tissue microarray are available in the Supplemental File.

### Supplemental Information

Supplemental information for this article can be found online at http://dx.doi.org/10.7717/peerj.11905#supplemental-information.

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
