# Peer review of "DOG1 is commonly expressed in pancreatic adenocarcinoma but unrelated to cancer aggressiveness"

_PeerJ, doi:10.7717/peerj.11905_

## Round 0.1 · original submission · Major Revisions

Dear Authors,

Your paper has now been evaluated by two reviewers that, although they found your paper with interest, feel that it needs major changes.

Please address all the points raised by the reviewers in your new version of the manuscript.

Thank you

Reviewer 1 ·

Basic reporting

Overall, the manuscript writing is clear and easy to understand. Also, the results are concise. With regard to reporting, I listed a few comments to further improve manuscript readability as seen below:

Line 72-73: this part is trying to summarize the reported cases of DOG1 across multiple human organs and cancer types, yet it is a bit unclear what exactly are the organ-cancer pairs given the current wording.

Line 80: "From data derived from the TCGA database". Recommend a rewrite. Also in line 80, do "these authors" refer to Crottès et al?

Line 86-90: Recommend to split into 2-3 sentences for better readability.

Experimental design

To account for confounding variables in the analysis, it might be better to add other metadata such as sex, age, smoking status, and also include the correlation test p-value (with pancreatic cancer).

Validity of the findings

Overall a straightforward study. A suggestion regarding the evidence supporting key finding: more details are needed to support the key result that "DOG1 gene expression is unrelated to pancreatic cancer aggressiveness". E.g. What are the exact clinico-pathological parameters used for the correlation test? Is it catogorical variable or continuous variable? Is the Chi-squred test applicable to the data types?

Additional comments

Cancer genomics/transcriptomics, and biomarker identification for pancreatic cancer are important topics, and we welcome the work from the authors to contribute. There are some areas outlined above that could be improved for better manuscript quality.

Reviewer 2 ·

Basic reporting

no comment

Experimental design

no comment

Validity of the findings

no comment

Additional comments

In this manuscript, the authors highlight the association of DOG-1 with pancreatic adenocarcinoma but not it's aggressiveness. The primary method of analyzing DOG-1 is immunohistochemistry of microarray samples followed by statistical analysis. The article is written in clear English. Tables and one figure is clear. Literature references seem to be in order. I have few general comments/questions-
1. Why did the authors not skip the non-informative samples completely? If only 501 out of 599 were analyzable then why not skip them? If the authors wish to include this anyhow, it is not a good idea to include this information as an independent result, especially not the first one.
2. Proof-reading for typos will help the manuscript.

---

## Round 0.2 · accepted · Accept

Thank you for addressing all the reviewers' concerns. Your manuscript is now acceptable for publication. Congratulations.

Reviewer 2 ·

Basic reporting

no comment

Experimental design

no comment

Validity of the findings

no comment

Additional comments

The authors have responded satisfactorily to my comment. Minor typos have also been addressed. I congratulate the authors on their work.